# Monitoring of Thermal and Moisture Processes in Various Types of External Historical Walls

**DOI:** 10.3390/ma13030505

**Published:** 2020-01-21

**Authors:** Dariusz Bajno, Lukasz Bednarz, Zygmunt Matkowski, Krzysztof Raszczuk

**Affiliations:** 1Faculty of Civil and Environmental Engineering and Architecture, UTP University of Science and Technology, Al. Prof. S. Kaliskiego 7, 85-796 Bydgoszcz, Poland; dariusz.bajno@utp.edu.pl; 2Faculty of Civil Engineering, Wroclaw University of Science and Technology, Wybrzeze Wyspianskiego 27, 50-370 Wroclaw, Poland; zygmunt.matkowski@pwr.edu.pl (Z.M.); krzysztof.raszczuk@pwr.edu.pl (K.R.)

**Keywords:** monitoring, historical masonry wall, hygrothermal processes, internal insulation

## Abstract

In order to create and make available the following: Design guidelines, recommendations for energy audits, data for analysis and simulation of the condition of masonry walls susceptible to biological corrosion, deterioration of comfort parameters in rooms, or deterioration of thermal resistance, the article analyzes various types of masonry wall structures occurring in and commonly used in historical buildings over the last 200 years. The summary is a list of results of particular types of masonry walls and their mutual comparison. On this basis, a procedure path has been proposed which is useful for monitoring heat loss, monitoring the moisture content of building partitions, and improving the hygrothermal comfort of rooms. The durability of such constructions has also been estimated and the impact on the condition of the buildings that have been preserved and are still in use today was assessed.

## 1. Introduction

Problems related to heat loss and the dampening of building partitions have always accompanied construction and in the most recent literature can be found in many works on this subject, e.g., regarding internal thermal insulation [1,2,3,4], thermal insulation of historical buildings [5,6,7,8], and even innovative insulation materials, 100% plant-based, allowing for the reduction of CO^2^ emissions in construction [9].

The preservation of buildings against soil moisture and condensation was and still is one of the main priorities of their builders, despite significant technological progress. As presented in [10,11,12,13], they are the most frequent causes of degradation of entire objects and low utility comfort.

The paper analyzes the technical condition of the external masonry walls of historical buildings with constructions known to be [14,15,16], constantly exposed to the influence of the external environment (UV radiation, insolation, precipitation, variable temperature, relative air humidity), as well as internal one (relative humidity of rooms and limited ventilation).

While monitoring hygrothermal processes, the measurement possibilities and the test results were analyzed, as presented in [17,18,19,20,21,22,23]. The conclusions of the work [24], in which the results of research on historical bricks were presented, were also used. Experimental data was compared with the results of standard procedures used to assess the thermo-physical behavior of historical masonry walls. The requirements set by the standard [25] were also taken into account.

When considering the subject of the additional thermal insulation of historical masonry walls from the inside aspects of this issue, presented in [26,27,28,29,30,31] have been taken into account. The problem of internal thermal insulation using a capillary active mineral plate or block systems is barely described in the literature, especially when it comes to the protection of external surfaces against moisture.

The paper discusses only part of the issues related to historical buildings, which are quite often ignored by people responsible for their technical condition, as they may seem to be of low importance. They concern dampness of building walls with moisture coming from outside and inside, resulting from physical processes taking place in them, which have direct influence on safety and durability of whole buildings. Not only the loss of load-bearing parameters of structures may lead to their destruction and even construction disasters, but also slow degradation due to processes taking place inside the walls. There are no two identical structures, therefore there can be no standard and universal methods of their maintenance and rescue.

Nowadays, quite often there is a need to carry out thermo-modernization of the buildings, but due to the historical value of the elevation it will not always be possible to do it on the external side of walls. Then, the other method, e.g., insulation from the inside, remains to be chosen. In this case, the method should be thoroughly verified by a calculation model and an “in situ” study, in terms of the possibility of negative effects that may appear even after several years of use, after its implementation. A very important element of such activities is the assessment of their impact on the durability, safety of the structure, functional safety of the facility, and the internal microclimate.

The authors of the paper have focused on selected research techniques and computational simulations in the field of moisture transfer (coming from various sources) inside building walls. These measures should lead to the development of reliable assessments of the technical condition of historical buildings, with particular attention paid to the elements and structures directly exposed, i.e., vertical walls. At the same time, they should confirm or not, the effectiveness of the designed solutions and their subsequent implementations. Simulation calculation models should cover longer, several-year periods (it is recommended that these periods should be at least 10 years) of prognosis of the impact of the implementation of the above methods on the safety and durability of these buildings and their components. In the further part of the paper, the authors have limited their attention to selected problems of appearance and migration of moisture in the walls of historical buildings.

The research methodology was proposed below, in Figure 1 describing the monitoring and intervention procedure. Research was based on several selected representative types of historical masonry walls (being a part of public buildings, located in southern Poland) for which material characteristics of particular masonry elements were presented.

The results of moisture measurements (made in several masonry walls of this type) at the turn of the last dozen or so years were presented. In the next step, simulation calculations were made on the basis of which the exploitation evaluation of external masonry walls, made in different technologies and in a different period of time, was made. The possibility of interference in order to improve the hygrothermal comfort of rooms was also proposed.

## 2. Materials and Monitoring Methods

Nowadays, there are still many unique old constructions, with the unusual shaping of external partitions of buildings and structures in relation to current requirements. A present-day engineer, who has a lot of possibilities of computer-aided design, would highly appreciate them. The structures of historic buildings are distinguished by their age, technology depending on the level of technical knowledge available at the time of construction and the type of building materials used. One of the basic building materials used in constructions in the past was the material easiest to obtain in a given area. With the passage of time, as a result of the development of trade, the applied technologies and building materials were disseminated to other regions. Some of the first building materials used were clay, wood, and stone. These were followed by ceramics (dried and fired bricks, terracotta, etc.), air and hydraulic binders, cast iron, iron (steel), concrete, ferroconcrete, composites.

The main component of masonry structures was and still is stone and brick. While stone was the cheapest material obtained in the immediate vicinity of the works, a brick was already a masonry element produced artificially, through various clay processing technologies, i.e., its drying and later firing. Stone, as a masonry element, had irregular shapes (even when it was carved up), while bricks from the very beginning were given into the shape of a cuboid, or wedge-shaped pieces with appearance and size adapted to the needs of execution and the shape of the structure. Bricks were formed from clay, lime, and sand, and in times closer to ours, bricks were produced and still are produced on the basis of clay. Both stone and bricks were used to make the foundations and walls of buildings, vaults, tanks, chimneys, aqueducts, viaducts, etc., and in the nineteenth century, they also became one of the two basic components of ceramic-steel ceilings of the sectional type and reinforced flat slabs such as the Klein type.

Elements of stone, brick and mixed brick, and stone masonry walls with the following material parameters were used in the research:-Limestone, unit density *ρ* = 2.68 g/cm^3^, open porosity *p* = 12.80%,-Solid brick, unit density *ρ* = 1.90 g/cm^3^, open porosity *p* = 22.30%,-Hollow brick, unit density *ρ* = 1.40 g/cm^3^, open porosity *p* = 22.30%.

The physical properties of the bricks and stones were adopted on the basis of numerous tests performed by the authors on the basis of current technical standards (taking into account the quantity and quality of samples, testing methods, and the method of calculating the value). The presented values are average values obtained from laboratory (destructive) and in situ (NDT) measurements performed in these and other historical objects, including the region (southern Poland). The obtained measurement results were each time confronted with other results available in the literature.

To monitor the mass moisture of masonry walls, the traditional drying and weighting method is used (also in the version using an automatic moisture analyzer) and indirect methods consisting in measuring the selected physical or chemical properties of the material. The division of these methods is presented in Table 1.

After the analysis of methods used for monitoring humidity processes in the walls presented in [23,32,33,34,35,36] and in Table 1, it was decided that the measurements of the state and scope of humidity of wall elements would be studied by one of the most accurate methods, i.e., the drying and weighting method. To verify the results, the nondestructive method based on the measurements of the dielectric properties of the material was used. The drying and weighting method was used to determine the distribution of moisture on the thickness of the wall and to scale electric meters. Samples for mass moisture determination were taken by drilling. The taken samples were tested on an automatic moisture analyzer. The mass moisture content was determined from Equation (1):(1)$$u_{m}=\frac{m_{w}-m_{s}}{m_{s}}\times 100\%$$
where:

*u_m_*—mass humidity (%),

*m_w_*—wet sample mass (g),

*m_s_*—dry sample mass (g).

Considering that gauge reading–weight moisture content correlations depend on the material’s other properties such as its chemical composition, porosity, porosity structure and the kind and concentration of the salts, the commonly used gauges require calibration.

The tests of mass humidity were carried out on four different masonry walls made of solid brick (Figure 2).

In building “A” the first research on the moisture content of walls was carried out in 1997. Next, horizontal damp-proof insulation was made (by chemical barrier–mineral injection). The measurements were repeated in 2004 and then after flooding the cellar interior with water (as a result of a failure of the sewage system) in 2018 (more than 20 years after the first measurement). In building “B” humidity measurements were carried out in 2007. Next, moisture horizontal insulations (by chemical barrier–mineral injection) were made. The measurements were repeated in 2018. In building “C” the measurements were made in 2009. The measurements were repeated in 2019. Anti-humidity and waterproof insulation were not performed in this building. In building “D” the measurements were made in 2016. Next, horizontal damp-proof insulation (by chemical barrier–mineral injection) and internal insulation with a thickness of 5 cm were performed. The measurements were repeated in 2019.

The results of mass moisture tests for these four load-bearing masonry walls are shown in Figure 3. In all cases in which the anti-humidity insulation was made, a significant (about 10%) decrease in mass moisture of the walls was observed. In the absence of such insulation, the mass humidity increased slightly during the 10 years of measurements.

## 3. Hygrothermal Calculations

Old historical buildings did not have any moisture or waterproof insulation. Nowadays, such objects are often waterproofed. They are revitalized and thermo-modernized. Revitalization and thermo-modernization are elements of the construction process, which should take into account the conditions of subsequent exploitation of the facilities. Each practical contact with historical buildings and monuments enriches the experience, but does not allow creating a universal model that comprehensively covers all the problems related to their safe maintenance or ensuring their durability. In order to be able to develop guidelines for this type of activities, thermal and humidity calculations were made with the assumption of the required amount of air exchange in the interior of buildings according to [36]. As can be seen, when analysing the data from Table 2, if this condition is not met, the results of the calculations in Table 3 will not even be close to reality. Calculations described in more detail in [4] were carried out for a homogeneous 38 cm thick wall (Figure 3, wall No. 4). It can be clearly observed here that in the case of a decrease in the level of air exchange in a room, the amount of moisture remaining permanently in the partition significantly increases.

Models of each analysed type of masonry wall occurring in southern Poland (including those in areas belonging to Germany before World War II) are presented in Figure 4.

An attempt was made to make calculations for as many representative types of walls as could currently be found in historical buildings, as well as those erected in the last 200 years. The following types of walls were adopted:

1. Stone wall 80 cm thick. Walls of this type were made of field or lime stone (not treated or cut up), merged with lime mortar or lime mortar with the cement addition (from the late nineteenth century). A field stone is obtained from the fields (irregular, unworked, without clear petrographic features). In southern Poland it is usually granite left by retreating ice or sandstone.

2. Brick and stone or stone layered wall 62 cm thick. A wall such as wall No. 1, but very often the face of these walls was made of bricks.

3. Layered brick wall with air void (with vertical lining) 62 cm thick. A rarely occurring wall structure, due to its low strength and low thermal parameters. It consisted of two external facing walls, mainly 12 cm thick each, tied with stiffeners of the same thickness perpendicular to them, i.e., 12 cm (at the entire height of the joined layers).

4. Solid brick wall 38 cm thick. A very often used type of wall in residential and public buildings. The walls were made mainly of solid bricks with lime mortar, cement, and lime mortar, less frequently cement bricks.

5. Solid brick wall 12 + 25 cm thick with a 6 cm gap. A similar model of a wall to the one described above (No. 4), separated by an internal air gap.

6. Solid brick wall with a single layer of perforated brick on the inside. A solid brick wall finished on the inside with a single layer of perforated brick to limit heat loss and to allow easier removal of moisture accumulated in it.

7. Solid brick wall with double layer of hollow bricks on the inside. A solid brick wall finished on the inside with a double layer of hollow brick to reduce heat loss and to allow for easier removal of moisture accumulated in it.

8. Solid brick wall approx. 70 cm thick. A very often used type of wall in the ground level of large objects. The walls were made mainly of solid bricks of lime mortar, cement, and lime mortar.

The current regulations [25], in Annex 2, let us assume the required critical value of the temperature factor *f_Rsi_* = 0.72 in rooms heated to the temperature of at least 20 °C, in residential buildings, collective residence, assuming that the average monthly relative humidity of indoor air is equal to *φ* = 50%.

The accepted calculation parameters and results are presented in Table 3.

Practically all partitions listed in Table 3 do not meet the standard [25], (results in column coefficient of heat transfer *U*), while detailed calculations performed with WUFI 2D v.3.4 [37] showed that surface moisture condensation will occur on the inner surfaces of some of the partitions (the possibility of surface condensation). Table 4 presents diagrams describing the amount of moisture in partitions in a 12-month cycle, in a 10-year period, counted from 2017.

In each of the analyzed partitions, the moisture level stabilizes after about 4–5 years of operation, assuming proper replacement of used and damp indoor air. The service life of partitions 5, 6, 7 during the first 4–5 years results in the increase of moisture inside them. None of the partitions meet the current requirements in terms of thermal insulation.

## 4. Monitoring and Hygrothermal Calculations of Masonry Adapted to New Conditions of Use

In order to meet modern guidelines, especially concerning the heat consumption of masonry walls in historical buildings, one of the best technical solutions seems to be the use of internal thermal insulation systems using capillary active mineral plaster, plasterboards, or blocks. Thanks to their very porous structure and very low density (*ρ* < 150 kg/m^3^), these systems achieve high thermal insulation parameters. The thermal conductivity coefficient (*λ* ≤ 0.05 W/m·K) and the diffusion resistance coefficient of water vapour (*µ* = approximately 5) are very low.

For a detailed analysis of the problem, one type of wall, a solid brick masonry wall of approximately 70 cm thickness (Table 3, No. 8), in which a horizontal damp-proof insulation was made and an additional insulation was applied from the inside with blocks of capillary active mineral layer 10 cm thickness. For bricklaying the blocks from the capillary active mineral layer was used as a systemic lightweight capillary active mortar with heat conduction coefficient *λ* equal to 0.2 W/m·K. Moisture monitoring was performed by means of 2 FP probes of the FOM2 system (Figure 5a), which were placed in the mortar layer between the original wall and the internal insulation layer (Figure 5b). The research was conducted continuously from November 2016 to September 2019.

The FOM2 system used for measurements is a set consisting of FOM2/mts measuring device and FP/mts probes operating on the basis of TDR (time domain reflectometry) and electrical conductivity. The measurement results can be sent to mobile devices thanks to the use of a Bluetooth connection. Measurement parameters of the FOM2 device are presented in Table 5.

Probe No. 1 was placed on the flat surface of the wall and probe No. 2 in the corner at a height of approximately 1.2 m above the floor level. Both probes were connected with the measuring device FOM2/mts. Before the probe was placed in the wall, it was calibrated using information on the length of cable connections and two dielectric standards of reference: Air and water. After calibration in the air, the probe should be placed in water (preferably pure and deionized at a temperature of about 20 °C) and recalibrated. Only by carrying out the calibration process can you be sure of the results obtained.

## 5. Results and Discussion

The results of the humidity measurements together with a diagram of the probe location in the tested masonry wall are presented in Figure 6. The results of temperature measurement (probe No. 1, probe No. 2, indoor temperature, outdoor temperature) are presented in Figure 7.

Monitoring of hygrothermal parameters was performed by analyzing the data obtained from humidity and temperature measurements and material parameters using WUFI 2D v.3.4 software [37]. When analyzing the state and distribution of temperature isotherms and the adiabate of heat streams from the last 18 months (Figure 8) in the case of the corner of an uninsulated masonry wall and capillarily insulated from the inside with a capillary active mineral layer of 10 cm thickness, it can be stated that a significant decrease in the cooling of the corner is visible. A significant decrease in the water content of the insulated wall was also recorded (Figure 9).

Similar numerical measurements were made for other types of the most popular historical external masonry walls presented in Figure 3. All of them were insulated from the inside with 10–15 cm mineral material (depending on demand). In all cases the coefficient of heat transfer *U* reached the values acceptable for the current standards of heat and moisture comfort of rooms. The lack of possibility of surface condensation was also observed.

The results obtained, apart from confirming the usefulness of internal thermal insulations performed with mineral systems, indicate one very important problem. In the case of insulation of walls from the inside, it is important to remember the proper protection of external wall surfaces. The paper [39] presents the results of the research which clearly show that additional insulation of historical masonry walls from the inside cannot be performed in walls without additional protection of their external surfaces against precipitation. In particular, if the masonry walls are exposed to acid rain corrosion worsens the mechanical properties of brick masonry. Otherwise, despite the internal insulation, it is possible to destroy the facade and see mould develop on the surfaces behind the internal insulation. Measurements and observations presented in [39,40,41] confirm these theses.

Thermal insulation of external partitions of buildings from the inside each time requires detailed hygrothermal analyses, because this method should not be treated as a universal one. Improper application of this method may lead to a faster technical wear and tear of partitions or even to their total degradation.

## 6. Conclusions

Walls are vertical partitions of buildings, which have been improved for many years not only to provide them with the required load-bearing capacity, but also to provide them with adequate protection against the external environment. The article, giving the example of a few selected external partitions of buildings, recalls the solutions applied over the last 200 years and a short history of their evolution to modern requirements. From a physical point of view, attention is drawn not only to the permanent problem of heat loss accompanying the building envelope, but also to the condensation and accumulation of moisture within it. The use of newer technologies did not solve this problem at all, and even showed how little is needed to damage such a structure the more complicated the structure is. New materials have improved the thermal insulation of partitions, but at the same time have made them more sensitive to both external and internal moisture, including condensation.

## Figures and Tables

**Figure 1 materials-13-00505-f001:**
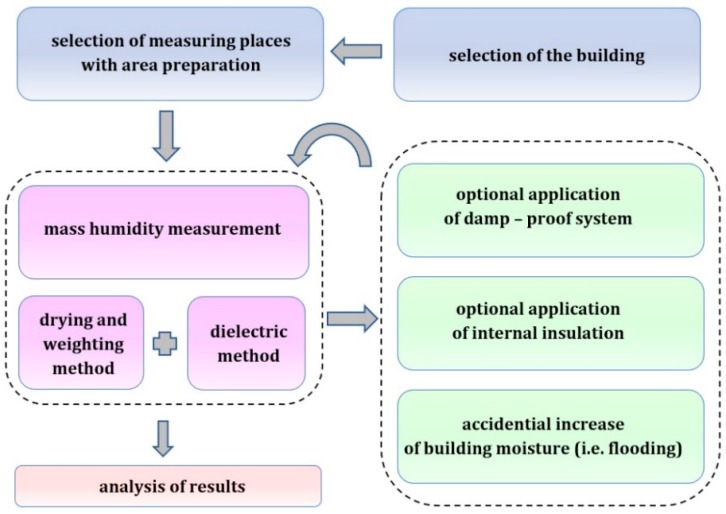
Monitoring and intervention procedure.

**Figure 2 materials-13-00505-f002:**
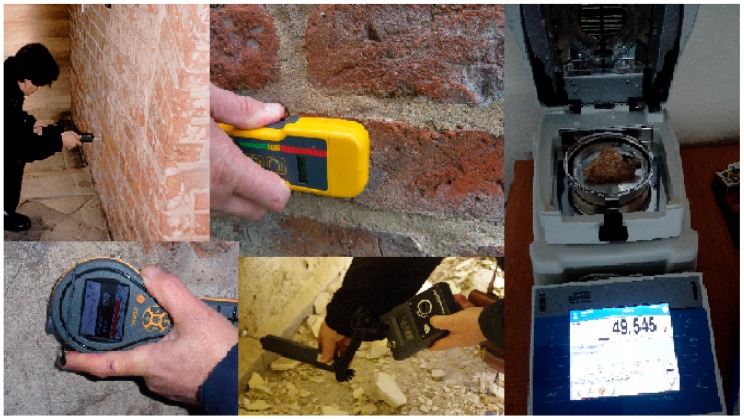
Testing areas and measurement techniques (nondestructive and destructive).

**Figure 3 materials-13-00505-f003:**
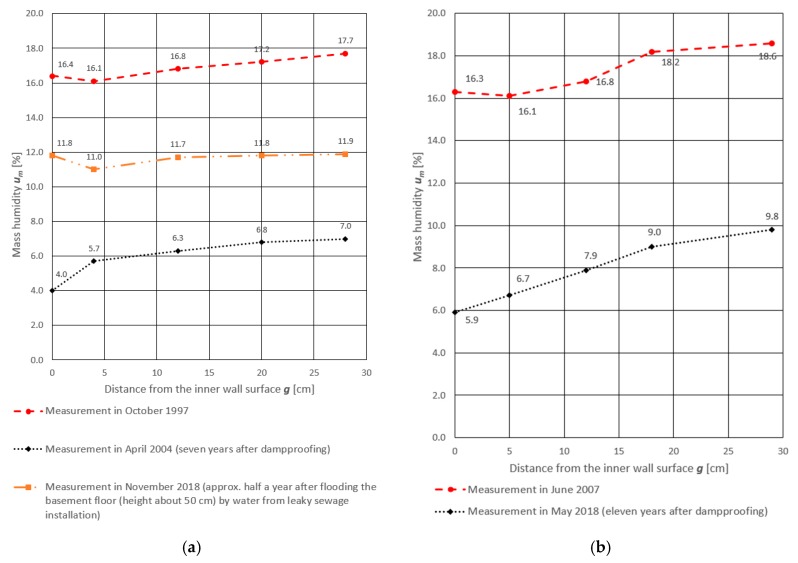
Results of mass moisture measurements on partition thickness in four different masonry walls made of solid brick: (**a**) Wall of building “A”; (**b**) wall of building “B”; (**c**) wall of building “C”; (**d**) wall of building “D”.

**Figure 4 materials-13-00505-f004:**
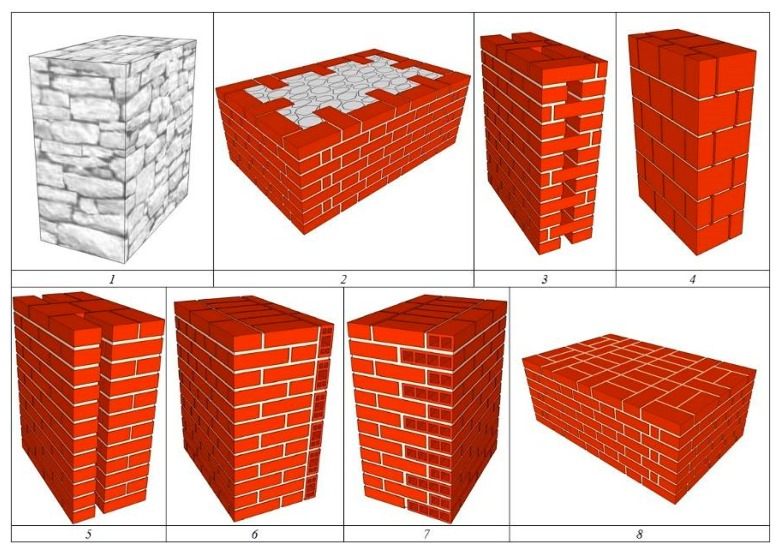
Models of the analyzed types of load-bearing and/or curtain masonry walls.

**Figure 5 materials-13-00505-f005:**
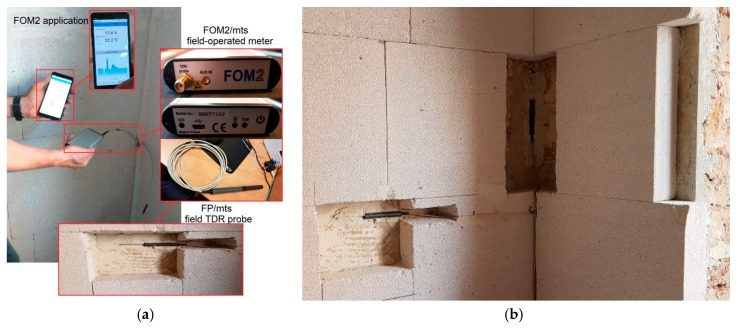
FOM2 measuring system: (**a**) Elements of the system; (**b**) probe No. 1 and probe No. 2 installed between the wall and the internal thermal insulation.

**Figure 6 materials-13-00505-f006:**
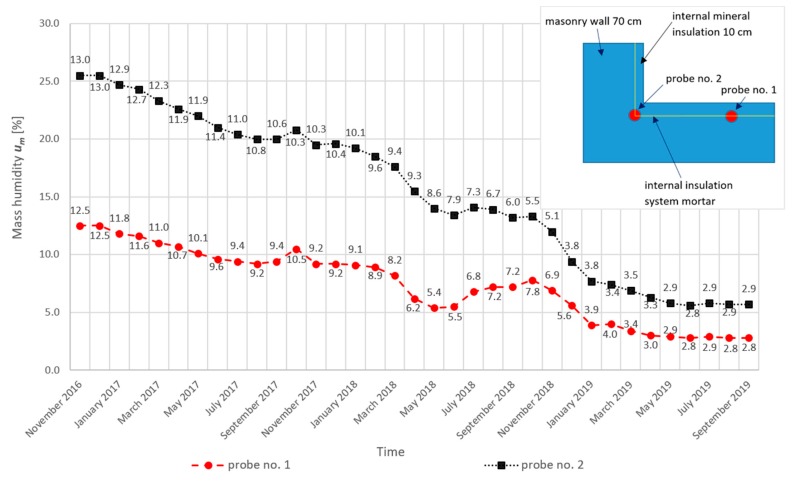
Results of humidity measurements together with a diagram of probes’ location in the tested wall.

**Figure 7 materials-13-00505-f007:**
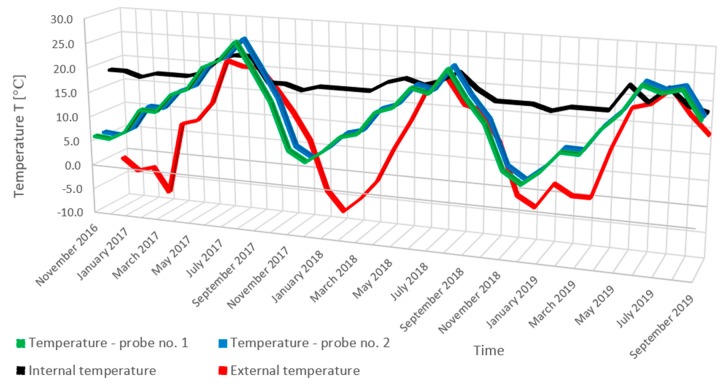
Temperature measurement results (probe No. 1, probe No. 2, indoor temperature, outdoor temperature).

**Figure 8 materials-13-00505-f008:**
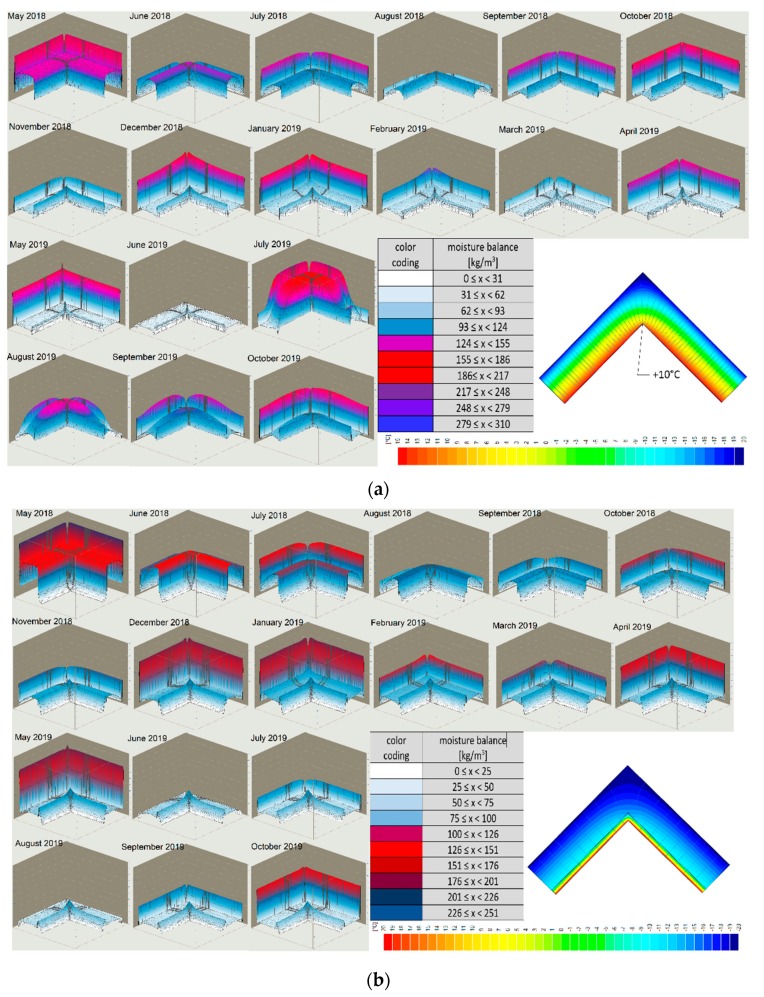
Isothermal distribution of temperature in the corner of the wall: (**a**) Uninsulated from the inside; (**b**) insulated from the inside.

**Figure 9 materials-13-00505-f009:**
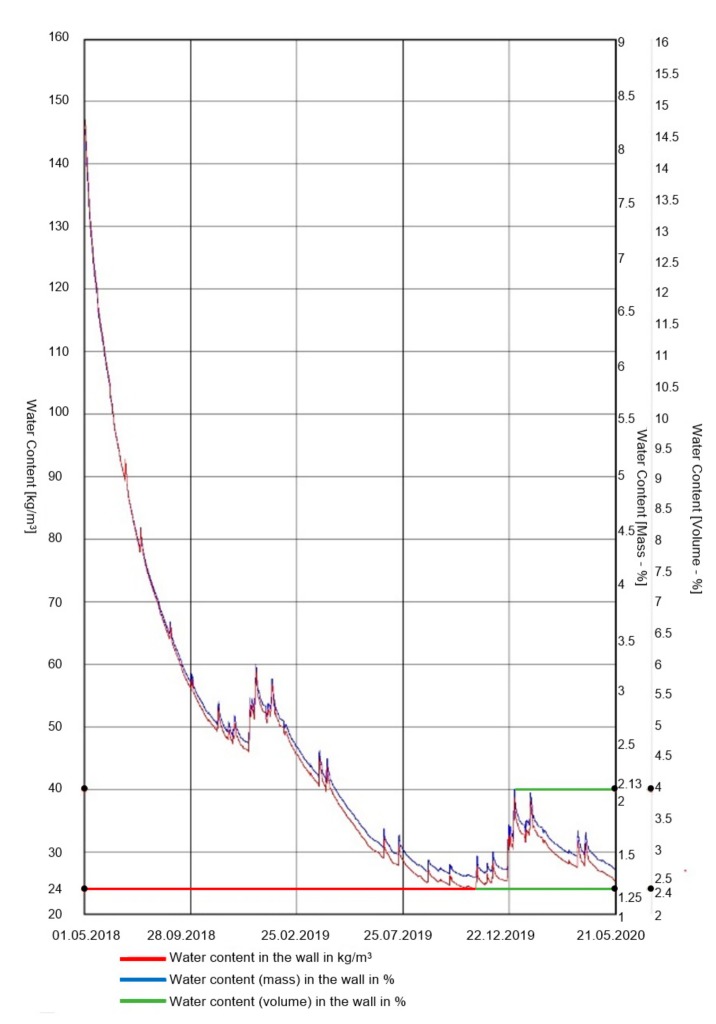
Water content in the insulated wall.

**Table 1 materials-13-00505-t001:** A classification of nondestructive methods determining moisture content in building materials.

Group of Methods	Name of Method	Measured Parameter
Chemical methods	1. indicator paper method	Change in indicator paper colour under the influence of damp material
	2. carbide method (CM)	Pressure of acetylene (formed from reaction of carbide with water) in a hermetic container
Physical electrical methods	1. electric resistance method	Change in material electric resistance as a result of change in dampness
	2. dielectric method	Change in material dielectric constant as a result of change in dampness
	3. microwave method	Attenuation of microwaves as they pass through the damp material
Physical nuclear methods	1. neutron method	Number of neutrons slowed down by collisions with hydrogen atoms
	2. gammascopy method	Change in γ radiation after it passes through investigated material

**Table 2 materials-13-00505-t002:** Dependence of the degree of barrier moisture on the air exchange rate.

No.	Air Exchange Rate per Hour	Moisture Level Increase during Central Heating Months	Increase of Moisture Retained within the Barrier
1	2	1	1
2	1	×2	×3
3	0.5	×3	×5

**Table 3 materials-13-00505-t003:** Wall cross sections with moisture level, distribution model, etc.

No.	Wall Cross Section	Moisture Level during Central Heating Months and Retained Moisturekg/m^3^	Moisture Distribution Model(Winter)	Coefficient of Heat Transfer*U*W/m^2^·K	Temperature Factor*f_Rsi_*	The Possibility of Surface CondensationX—yesV—no
1	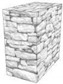	45.3/21.7	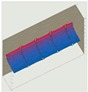	1.551	0.962	X
2	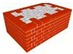	22.0/2.8	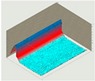	1.359	0.824	X
3	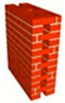	8.5/0.8	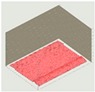	1.755	0.772	V
4	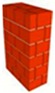	15.1/11.6	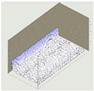	1.567	0.796	X
5	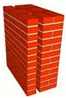	56.7/32.1	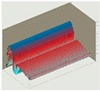	1.392	0.819	X
6	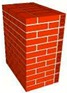	50.0/29.2	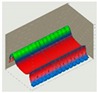	1.258	0.852	X
7	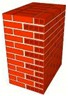	41.0/22.0	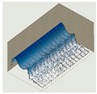	0.857	0.915	V
8	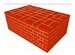	50.2/46.7	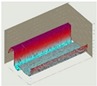	0.927	0.768	V

**Table 4 materials-13-00505-t004:** Dependence of the degree of barrier moisture on the air exchange rate.

No.	Graphs Showing Moisture Content in the Whole Barrier Cross Section over 10 Years in kg/m^3^
1	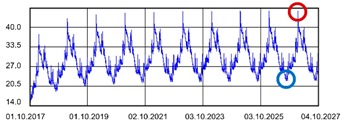
2	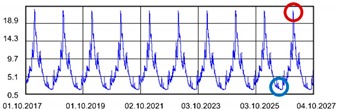
3	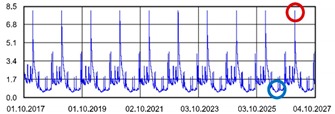
4	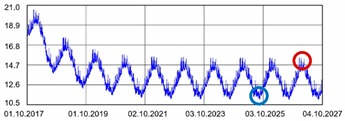
5	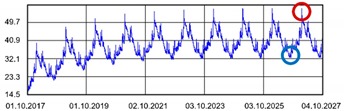
6	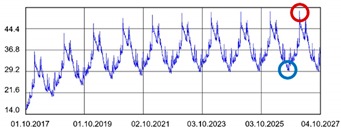
7	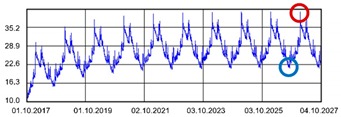
8	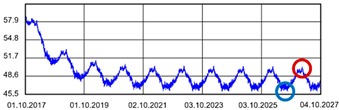

**Table 5 materials-13-00505-t005:** FOM2 specification according to [38].

Range of Volumetric Moisture %	Range of Temperature °C	Range of Electrical Conductivity S/m	Moisture Absolute Error %	Temperature Absolute Error °C	Electrical Conductivity Relative Error %
0–100	(−20)–(+50)	0.000–1	±2	±0.5	±10

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
