# Peer review of "Monitoring of Thermal and Moisture Processes in Various Types of External Historical Walls"

_materials, 2020, doi:10.3390/ma13030505_

Round 1
Reviewer 1 Report
The paper is interesting because focus on the evaluation of moisture and thermal processes of several historical walls. However, sometimes is confusing the relation between the different sections and procedures in each section. The authors should improve the structure of the paper (explaining the connection between each section) and highlight the main contribution of the paper. The following improvements are suggested:
The authors refer several references without highlighting the relevance from each reference to the present study. Especially lines 26, 35 and 41. Line 49 replace “ina” by “in a” Lines 76, 77, 78 replace cm3 by “cm3” The authors should refer the method used to determine the porosity. Open porosity? Some issues are not clear: the dimensions of the samples for the gravimetric method; the techniques from table 1 that the authors used in this study; Could the authors add some illustrations about the testing areas and techniques? The authors should clarify the testing areas: the number, the position on the wall; The external environmental conditions during the measurements should be added to the text in the different sections; The authors refer the anti-humidity procedure on the walls. Could you explain better the techniques for each wall type? The damp-proof insulation is a chemical or a physical barrier? The authors should clarify in which countries the selected walls are representative in historic buildings. For example, stone and brick rubble wall is a common solution but is not mentioned by the authors; In figure 2 please say if the walls are load bearing or non-load bearing walls Please insert the units in table 3 In lines 175-178 instead of giving the number of the columns, please give the title of the column for a better understanding Please improve the quality of figures in table 4 (the numbers are too small) Please explain the technology used in line 195 –active mineral layer of 10 cm thickness. A plate? Board? A reinforced plaster? Please relate the technique of figure 3 with the table 1; and also the studied wall solution Improve the quality of figure 7 Some disadvantages of applying the internal thermal on walls of historical are not discussed, for example the negative influence for the building thermal inertia. It is not clear in the text – line 259-260 “a short history of their evolution to modern technologies”. What are the walls included in modern technologies? The title focus only on historical walls.
Author Response
Detailed response to reviewers of the paper entitled “Monitoring of thermal and moisture processes in various types of external historical walls” prepared by Dariusz BAJNO, Łukasz BEDNARZ, Zygmunt MATKOWSKI and Krzysztof RASZCZUK.
The authors would like to thank the reviewers for agreeing to review our manuscript and particularly for very useful comments. The corrections are highlighted yellow in the text.
Responding to reviewer #1, we would like to state the following:
Specific comments:
Reviewer #1 comment/recommendation |
Status |
Our comments |
Sometimes is confusing the relation between the different sections and procedures in each section. The authors should improve the structure of the paper (explaining the connection between each section) and highlight the main contribution of the paper. |
Corrected and added in revised text |
The paper discusses only part of the issues related to historical buildings, which are quite often ignored by people responsible for their technical condition, as they may seem to be of low importance. They concern dampness of building walls with moisture coming from outside and inside, resulting from physical processes taking place in them, which have direct influence on safety and durability of whole buildings. Not only the loss of load-bearing parameters of structures may lead to their destruction and even construction disasters, but also slow degradation due to processes taking place inside the walls. The authors of the paper have focused on selected research techniques and computational simulations in the field of moisture transfer (coming from various sources) inside building walls. These measures should lead to the development of reliable assessments of the technical condition of historical buildings, with particular attention paid to the elements and structures directly exposed, i.e. vertical walls. At the same time they should confirm or not, the effectiveness of the designed solutions and their subsequent implementations. Simulation calculation models should cover longer, several-year periods (it is recommended that these periods should be at least 10 years) of prognosis of the impact of the implementation of the above methods on the safety and durability of these buildings and their components. In the further part of the paper, the authors have limited their attention to selected problems of appearance and migration of moisture in the walls of historical buildings. |
The authors refer several references without highlighting the relevance from each reference to the present study. Especially lines 26, 35 and 41. |
Corrected and added in revised text |
The authors cited only a few known and available literature titles. It was not the intention of the authors to exactly execute the state of the art of these papers. They only indicated that the topic of monitoring of thermal and moisture processes in various types of external historical masonry walls is quite interesting. Other authors also try to propose a procedure for monitoring heat loss and the moisture content. |
Line 49 replace “ina” by “in a” |
Corrected in revised text |
Thank you for the apt and valuable hints. |
Lines 76, 77, 78 replace cm3 by “cm3” |
Corrected in revised text |
Thank you for the apt and valuable hints. |
The authors should refer the method used to determine the porosity. Open porosity? |
Corrected and added in revised text |
Porosity was determined according to PN-EN 1936:2010 Natural stone test methods - determination of real density and apparent density, and of total and open porosity. Yes, it is open porosity. |
Some issues are not clear: the dimensions of the samples for the gravimetric method; the techniques from table 1 that the authors used in this study. |
Added in revised text |
For laboratory tests using the drying and weighting method, were used irregular samples (weighing up to 100 g) obtained from the tested walls. Physical dielectric method from Table 1 have been used for moisture non destructive testing (NDT). |
Could the authors add some illustrations about the testing areas and techniques? |
Added in revised text |
Figure 2 |
The authors should clarify the testing areas: the number, the position on the wall |
Unchanged |
The position on the wall for the tests regarding moisture content (section 2) is explained on figure 1, where the “x” axis represents the distance from the inner wall surface (cm) |
The external environmental conditions during the measurements should be added to the text in the different sections |
Unchanged |
The range of temperature is to find in Table 5. |
The authors refer the anti-humidity procedure on the walls. Could you explain better the techniques for each wall type? The damp-proof insulation is a chemical or a physical barrier? |
Added in revised text |
Horizontal insulation was performed by injection. This damp-proof insulation is a chemical barrier. |
The authors should clarify in which countries the selected walls are representative in historic buildings. For example, stone and brick rubble wall is a common solution but is not mentioned by the authors |
Added in revised text |
The walls are located in southern Poland. Buildings A,B,C,D are public buildings. All constructed in the XIXth century. Fragments of the studied walls may be earlier. Of course, the stone and brick rubble wall is a common structure, but it does not occur often in southern Poland and therefore it is not analysed by the authors. The authors will also try to analyse this type of wall in their next research papers. |
In figure 2 please say if the walls are load bearing or non-load bearing walls |
Added in revised text |
In Figure 2 (currently Figure 4) the walls are load-bearing and/or curtain walls. This was of little importance for the research carried out, but the thickness of these walls allowed them to perform both curtain and load-bearing functions. |
Please insert the units in table 3 |
Added in revised text |
Thank you for the apt and valuable hints. |
In lines 175-178 instead of giving the number of the columns, please give the title of the column for a better understanding |
Corrected in revised text |
Thank you for the apt and valuable hints. |
Please improve the quality of figures in table 4 (the numbers are too small) |
Corrected in revised text |
Thank you for the apt and valuable hints. |
Please explain the technology used in line 195 –active mineral layer of 10 cm thickness. A plate? Board? A reinforced plaster? |
Corrected in revised text |
The additional insulation was applied from the inside with blocks of capillary active mineral layer 10 cm thick. |
Please relate the technique of figure 3 with the table 1; and also the studied wall solution |
Paper structure unchanged |
There is no direct connection between figure 3 and table 1. During the moisture measurements, two methods were used: drying and weighting method and dielectric method. |
Some disadvantages of applying the internal thermal on walls of historical are not discussed, for example the negative influence for the building thermal inertia. |
Paper structure unchanged |
In situations where the insulation will be on the inside of the wall, the range of the negative temperature zone may cover the entire depth of the insulated element. Then the water accumulated in the pores and capillaries of the material, increasing its volume can lead to permanent frost damage. The external layers of partitions are particularly exposed to the destructive influence of frost, in which usually the most moisture (water vapour) accumulates, constantly replenished by precipitation. A very unfavourable influence on these partitions will be frequent temperature jumps from their positive to negative values. Much worse will be the situation when external expeditions (plasters) will be damaged (they will have cavities or will be cracked or lost), then in the partitions. which are to protect will increase the moisture level. The thermal resistance of the partitions will be identical, regardless of whether the insulation with the same parameters will be placed on its external or internal side. The situation connected with thermal inertia of partition will be different, which will translate into microclimate of compartments. While thermal insulation layer laid on external side of partitions will allow full use of wall material for heat accumulation and regulation of microclimate inside compartments (it concerns both winter and summer period), location of thermal insulation on internal side of partitions will not allow for heat accumulation in winter period, because insulated partition will be separated from interior of compartments by material with low thermal conductivity coefficient λ and simultaneously with low thermal capacity. For example, if the thermal capacity of solid ceramic brick masonry will be at the level of approx. 435 kJ/(m2K), then the thermal capacity of currently used thermal insulation materials, such as: mineral wool, glass wool, foamed polystyrene, Multipor mineral boards reached only 10÷20 kJ/(m2K), i.e. approx. 40 ÷ 20 times lower. Moreover, in situations described above, when the ends of wooden ceiling joists have their support on external walls, the growth of condensation moisture that will occur here will be conducive to the appearance and development of biological corrosion and, at the same time, the threat of building disasters. |
It is not clear in the text – line 259-260 “a short history of their evolution to modern technologies”. What are the walls included in modern technologies? The title focus only on historical walls. |
Corrected in revised text |
When the authors wrote "a short history of their evolution to modern technologies", they meant adapting historical building partitions to modern requirements. |
Reviewer 2 Report
The authors propose a new procedure for monitoring heat loss and the moisture content of building partitions with estimation of the durability conditions. Methodology is validated in several real buildings. The topic is quite interesting in terms of energy efficiency in buildings, so paper should be proposed for acceptance. Some minors changes are recommended to be included: 1) Please to improve the Introduction section, being more specific in the previous scientific developments; 2) Section 2: Please to include a flow chart describing the monitoring procedure. It would be useful for readers in order to apply the technique. 3) Quality of Figure 1 es very poor. 4) Figure 6, please to change the arrangement of Figures a) and b) in order to enlarge the figures. A vertical arrangement would be better than an horizontal one.
Author Response
Detailed response to reviewers of the paper entitled “Monitoring of thermal and moisture processes in various types of external historical walls” prepared by Dariusz BAJNO, Łukasz BEDNARZ, Zygmunt MATKOWSKI and Krzysztof RASZCZUK.
The authors would like to thank the reviewers for agreeing to review our manuscript and particularly for very useful comments. The corrections are highlighted yellow in the text.
Responding to reviewer #2, we would like to state the following:
Specific comments:
Reviewer #2 comment/recommendation |
Status |
Our comments |
Please to improve the Introduction section, being more specific in the previous scientific developments |
Added in revised text. |
Thank you for the apt and valuable hints. The Introduction section was improved. |
Section 2: Please to include a flow chart describing the monitoring procedure. It would be useful for readers in order to apply the technique |
Added in revised text. |
Thank you for the apt and valuable hints. A flow chart was added in revised text. |
Quality of Figure 1 es very poor. |
Corrected in revised text |
Thank you for the apt and valuable hints. Figure 1 (actually 3) was corrected. |
Figure 6, please to change the arrangement of Figures a) and b) in order to enlarge the figures. A vertical arrangement would be better than an horizontal one. |
Corrected in revised text |
Thank you for the apt and valuable hints. Figure 6 (actually 8) was changed. |
Reviewer 3 Report
The paper presents some methodological limitations:
1 - The main subject of study is masonry wall behavior but the masonry concept is not clearly identified. Sometimes the authors mentione "wall" "wall structures" instead of masonry.
2 - The authors mentione also that the main aim is to study walls in historic buildings. Neverthless nothing is said about were the buildings a, b, c and d are located. Are they located in a same country? In what region? Anything is said also about the historical period of their production.
3 - The paper refers
"Old historical buildings did not have any moisture or waterproof insulation. Nowadays, these objects are often waterproofed. They are revitalized and thermo-modernized"
This statement does not take into account that many masonry buildings were in the past covered with renderings and often limewahsed. Those coatings were given to the buildings some degee of water proof insulation. These reality can not be avoided when someone intends to study old masonry buildings.
4 - Renderings and coatings are not mentioned or studied in the paper.
5 - The composition of mortars used to bound the masonry blocks are not mentioned either.
6 - Physical characteristics of bricks are established without criteria. In reality, whend someone study an historical building must find an average of physical characteristics of blocks and mortars. A representative number of samples are studied and then some average characteristics are assumed to analyse the building.
7 - In the section "Hygrothermal calculations" different models of masory are used. Unfortunately, the majority of those models do not refer to "historic masonries". Until the beggining of the 20th century hollow or perfurated bricks were not used.
The authors refer also on line "145": "Stone wall 80 cm thick. Walls of this type were made of field or lime stone". This type of description is not at all accurated. "Field stone" means only that the stone was recovered from the subsoil or the topsoil of a field. It does not say anything about the petrographical charcateristics of the stone: in a region A we can find a fieldstone which is a limestone. In another region B, the field stone can be a granit, for example.
8 - The paper lacks of methodology linked with preservation of historic buildins.
9- Any distinction is made between laying/beding mortars and rendering/coating mortars.
10- The chemical composition of mortars and the physical contribution of mortars to the behaviour of a masonry wall is not clearly taken in consideration.
Author Response
Detailed response to reviewers of the paper entitled “Monitoring of thermal and moisture processes in various types of external historical walls” prepared by Dariusz BAJNO, Łukasz BEDNARZ, Zygmunt MATKOWSKI and Krzysztof RASZCZUK.
The authors would like to thank the reviewers for agreeing to review our manuscript and particularly for very useful comments. The corrections are highlighted yellow in the text.
Responding to reviewer #3, we would like to state the following:
Specific comments:
Reviewer #3 comment/recommendation |
Status |
Our comments |
The main subject of study is masonry wall behavior but the masonry concept is not clearly identified. Sometimes the authors mentione "wall" "wall structures" instead of masonry. |
Corrected in revised text |
The main aim for using the term: “masonry” in the paper was to underline a type of building material. In case of using term: “wall”, the goal was to underline the type of tested element in the building. |
The authors mentione also that the main aim is to study walls in historic buildings. Neverthless nothing is said about were the buildings a, b, c and d are located. Are they located in a same country? In what region? Anything is said also about the historical period of their production. |
Corrected in revised text |
Buildings “A”,”B”,”C”,”D” are public buildings. The walls were parts of buildings built in the XIXth century but fragments of the studied walls may be earlier, from different historical periods. |
The paper refers "Old historical buildings did not have any moisture or waterproof insulation. Nowadays, these objects are often waterproofed. They are revitalized and thermo-modernized". This statement does not take into account that many masonry buildings were in the past covered with renderings and often limewahsed. Those coatings were given to the buildings some degee of water proof insulation. These reality can not be avoided when someone intends to study old masonry buildings. |
Corrected in revised text |
Old historical buildings did not have any horizontal waterproofing. The existing vertical waterproofing was often made of clay and was ineffective when water conditions in the ground changed. In practice, the role of building partition coating is very important (apart from decorative one), which is very often forgotten. They protect external elements of building structures against destructive influence of environment. Therefore, cracks and losses of coating may lead to partial or total degradation of building materials with low resistance to daily and seasonal temperature fluctuations, strong sunlight, wind, UV rays, precipitation. On the one hand, they may limit the diffusion of water vapour in the partitions, on the other hand, they will protect them from excessive "supply" of new portions of sorption and precipitation moisture, while protecting them against UV radiation, wind and mechanical damage. They do not play a significant role in heat exchange through building partitions. They play a particularly valuable role in historical buildings by preventing mechanical losses of lime-based joints and washing out of hydraulic binder from them. Their role as the insulation of elements recessed in the ground is of secondary importance. This applies mainly to older lime-based expeditions, whose diffusion resistance was very low. Below, a few building materials and the water vapour diffusion resistance coefficient µ ( (moisture absorption/release of moisture) characterizing them are listed: mineral wool - µ=1; polystyrene foam µ=60÷80, ceramic, silicate and cellular concrete masonry elements µ=5/20, concrete 70/150, bitumen insulations and films µ=10,000÷100,000, cement plaster µ=25, cement and lime plaster µ=19, lime plaster µ=7. Thus, lime plaster (all the more old) had very little resistance to inflow water, let alone hydrostatic pressure. Moreover, the resistance of typical lime mortars, e.g. on walls recessed in the ground, is too low to effectively and permanently protect these partitions from the inflow of moisture. |
Renderings and coatings are not mentioned or studied in the paper. |
Paper structure unchanged |
The article does not analyse the participation of renderings and coatings in the processes taking place in masonry walls. Paper is focused only on the doping from the inside with blocks of capillary active mineral layer 10 cm thick. |
The composition of mortars used to bound the masonry blocks are not mentioned either. |
Added in revised text. |
For bricklaying the blocks from capillary active mineral layer was used systemic lightweight capillary active mortar with heat conduction coefficient λ equal to 0,2 W/m·K. |
Physical characteristics of bricks are established without criteria. In reality, when someone study an historical building must find an average of physical characteristics of blocks and mortars. A representative number of samples are studied and then some average characteristics are assumed to analyse the building. |
Added in revised text. |
The physical properties of the bricks and stones were adopted on the basis of numerous tests performed by the authors on the basis of current technical standards (taking into account the quantity and quality of samples, testing methods and the method of calculating the value). The presented values are average values obtained from laboratory (destructive) and in situ (NDT) measurements performed in these and other historical objects, including the region (southern Poland). The obtained measurement results were each time confronted with other results available in the literature. |
In the section "Hygrothermal calculations" different models of masory are used. Unfortunately, the majority of those models do not refer to "historic masonries". Until the beggining of the 20th century hollow or perfurated bricks were not used. |
Added in revised text |
Various models of walls occurring in southern Poland (including those in areas belonging to Germany before World War II) were studied. According to the authors, the walls from 150 or 200 years ago are historical walls. Due to their location on the map of Europe, these areas were a field of frequent fighting and wars. We do not have many historical objects from 200 years ago and more, as in southern Europe (Italy, Spain, Portugal, etc). The question is: which walls are historical and which are not? Hollow or perforated bricks started to be made in the second half of the XIXth century. In the Prussian Empire and the Austro-Hungarian Empire (present-day southern Poland) the walls with hollow or perforated bricks started to be made in the late XIXth century. According to the authors, these are historical constructions. Such information can be found e.g. in XIXth century literature, such as Menzel C.A: Der Steinbau. Handbuch fuer Architekten, Bauhandwerker and Bauschuler. Karlsruhe 1885. |
The authors refer also on line "145": "Stone wall 80 cm thick. Walls of this type were made of field or lime stone". This type of description is not at all accurated. "Field stone" means only that the stone was recovered from the subsoil or the topsoil of a field. It does not say anything about the petrographical charcateristics of the stone: in a region A we can find a fieldstone which is a limestone. In another region B, the field stone can be a granit, for example. |
Corrected in revised text |
The authors, when using the term "field stone", meant a stone obtained from the fields (irregular, unworked, without clear petrographic features). In southern Poland it is usually granite left by retreating ice or sandstone. |
The paper lacks of methodology linked with preservation of historic buildins. |
Included in revised text |
The methodology related to the study of the preservation of historical buildings has been presented in many publications. A part of the methodology of conservation and thermo-modernization of historical masonry walls related to thermal and moisture processes the authors tried to present in Figure 1. |
Any distinction is made between laying / beding mortars and rendering/coating mortars. |
Paper structure unchanged |
The article does not analyse the participation of renderings and coatings mortars and laying / bedding mortars in the processes taking place in masonry walls. In the case of historical walls from southern Poland these were almost the same mortars based on lime and sand. |
The chemical composition of mortars and the physical contribution of mortars to the behaviour of a masonry wall is not clearly taken in consideration. |
Paper structure unchanged |
The article does not analyse the chemical composition of mortars and the physical contribution of mortars to the behaviour of a masonry wall; data used to calculate the migration of moisture in historical walls (bricks and historical mortars) were from own research and research made in Fraunhofer Institute fur Bauphysik IBP in Stuttgart, Germany and Technical University of Vienna, Austria. |